# Dual Deep Sequencing Improves the Accuracy of Low-Frequency Somatic Mutation Detection in Cancer Gene Panel Testing

**DOI:** 10.3390/ijms21103530

**Published:** 2020-05-16

**Authors:** Hiroki Ura, Sumihito Togi, Yo Niida

**Affiliations:** 1Center for Clinical Genomics, Kanazawa Medical University Hospital, 1-1 Daigaku, Uchinada, Kahoku, Ishikawa 920-0923, Japan; togi@kanazawa-med.ac.jp (S.T.); niida@kanazawa-med.ac.jp (Y.N.); 2Division of Genomic Medicine, Department of Advanced Medicine, Medical Research Institute, Kanazawa Medical University, 1-1 Daigaku, Uchinada, Kahoku, Ishikawa 920-0923, Japan

**Keywords:** somatic variant detection, clinical sequencing, next generation sequencing, cancer gene panel testing, somatic variant caller, mosaic mutation

## Abstract

Cancer gene panel testing requires accurate detection of somatic mosaic mutations, as the test sample consists of a mixture of cancer cells and normal cells; each minor clone in the tumor also has different somatic mutations. Several studies have shown that the different types of software used for variant calling for next generation sequencing (NGS) can detect low-frequency somatic mutations. However, the accuracy of these somatic variant callers is unknown. We performed cancer gene panel testing in duplicate experiments using three different high-fidelity DNA polymerases in pre-capture amplification steps and analyzed by three different variant callers, Strelka2, Mutect2, and LoFreq. We selected six somatic variants that were detected in both experiments with more than two polymerases and by at least one variant caller. Among them, five single nucleotide variants were verified by CEL nuclease-mediated heteroduplex incision with polyacrylamide gel electrophoresis and silver staining (CHIPS) and Sanger sequencing. In silico analysis indicated that the *FBXW7* and *MAP3K1* missense mutations cause damage at the protein level. Comparing three somatic variant callers, we found that Strelka2 detected more variants than Mutect2 and LoFreq. We conclude that dual sequencing with Strelka2 analysis is useful for detection of accurate somatic mutations in cancer gene panel testing.

## 1. Introduction

Next generation sequencing (NGS) is a powerful technology used in clinical genetic testing for the diagnosis of cancer and inherited diseases [1,2,3]. Whole genome sequencing or whole exome sequencing are excellent tools for the comprehensive detection of somatic and germ-line mutations [4]. However, information about mutations identified using these techniques is not always applied effectively in clinical practice involving cancer because many mutations are difficult to interpret in association with cancer pathogenesis. Cancer gene panel testing assesses hundreds of genes, in which only clinically important genes are included in order to detect “actionable” mutations for clinical diagnostics [5]. At present, cancer gene panel testing is the first line of examination for clinical cancer diagnosis and is a time-saving and cost-effective method involved in cancer treatment.

Accumulation of somatic mutations, such as single nucleotide variants (SNVs) and insertion and deletion variants (INDELs) disrupt cellular homeostasis and result in initiation and progression of cancer. Somatic mutations in cancer are mosaic, unlike germline mutations that are present in all cells of the body [6]. The samples used for clinical examination, whether surgically resected tissue or biopsy specimens, contain not only tumor cells but also normal stromal cells. Accordingly, cancer gene panel testing requires accurate detection of rare somatic variants. However, the innate errors in DNA sequencing methods, including errors during DNA amplification, NGS sequencing, and somatic variant calling, make analysis difficult. It is difficult to distinguish between real low-frequency somatic variants and artifacts caused by DNA sequencing methods. During the DNA amplification step of NGS library preparation, de novo low-frequency nucleotide sequence alternations are generated by DNA polymerase errors, even when using high-fidelity enzymes to amplify the targets [7,8,9]. Moreover, PCR polymerase errors affect not only the detection of low-frequency variants but also accurate measurement of allele frequency [10,11,12,13]. Previous studies have compared the error rates of several DNA polymerases for generating heterozygous and homozygous variants but there have been no studies that have examined the detection and measurement of allele frequency for de novo low-frequency variants [14].

The mathematical algorithms used by somatic variant callers are diverse [15]. Strelka2 is a recently developed software that has been shown to outperform the current somatic variant callers in both accuracy and computing cost [16]. Mutect2 in the Genome Analysis Toolkit (GATK) developed by the Broad Institute is a commonly used somatic variant caller [17]. LoFreq is a fast and sensitive somatic variant caller that uses full base-call qualities and other sources of errors inherent in sequencing, such as alignment uncertainty [18]. There are several studies on the performance of multiple somatic variant callers and the results indicate that Strelka2, Mutect2, and LoFreq are all relatively accurate and reliable compared to other somatic variant callers [19,20,21,22].

At present, no studies to our knowledge have compared DNA polymerases and somatic variant callers. In this study, we compared the performance of three high-fidelity DNA polymerases and three types of somatic variant caller software that were reported as most reliable for single tumor samples. We performed dual deep sequencing to investigate the reproducibility of each step, such as DNA amplification and variant calling. We also compared the performance of coverage and mutation detection abilities of each variant caller in several different numbers of mapped reads. We provide a practical NGS-based somatic variant calling workflow applicable to cancer gene panel testing, which is advantageous for more accurate somatic mutation detection.

## 2. Results

To examine the accuracy of somatic mosaic variant detection by NGS, we performed the workflow shown in Figure 1. In the pre-amplification (pre-capture) step of the library preparation, we used three different DNA polymerases to compare the fidelity. We also compared the number of detected somatic variants between three different somatic variant callers (Strelka2, Mutect2, and LoFreq). We validated the accuracy of the detected somatic variants using CHIPS technology and Sanger sequencing.

### 2.1. Comparison of Mapping Status between Dual Deep Sequencing

We compared the differences in mapping status between different polymerases. The number of reads mapped to the whole genome using the KAPA and NEB polymerases were higher than that of the Agilent polymerase in both blood and tumor samples, although the same amount of DNA was used for pre-amplification by these three polymerases (Figure 2A). The average depth on target region in the second deep sequencing was more than that of the first deep sequencing, indicating that mapped reads on target regions was greater in the second deep sequencing. The average depth using the KAPA and NEB polymerases was greater than that of Agilent in both deep sequencing experiments (Figure 2B). The average coverage rate was greater using the KAPA and NEB polymerases than the Agilent polymerase, in both blood and tumor samples (Figure 2C). The percentage of 100% coverage target regions was almost same in both deep sequencing experiments, with higher levels using the KAPA and NEB polymerases than the Agilent polymerase (Figure 2D). On the other hand, the percentage of 0% coverage target regions in the second deep sequencing is lower than in the first deep sequencing, with lower levels in both KAPA and NEB as compared to Agilent (Figure 2E). These results indicate that the mapping status of the target depth and coverage is variable from experiment to experiment, independent from the total number of mapped reads. Also, these findings indicate that variations in the homogeneity of target amplification by PCR during library preparation between experiments may affect mapping status. KAPA and NEB DNA polymerases were better than the Agilent DNA polymerase for efficiency and uniformity of amplification.

### 2.2. Comparison of Somatic Variants between Dual Deep Sequencing

We next examined the accuracy and the detection ability of three somatic variant callers (Strelka2, Mutect2, and LoFreq). About 2000 to 4000 somatic variants (about 300 to 700 filtered variants) were detected by Strelka2. Mutect2 detected about 200 to 1000 variants in the first deep sequencing experiment, but about 1700 to 3000 variants in the second deep sequencing (about 100 to 500 filtered variants in first and about 500 to 1000 filtered variants in second). LoFreq calls only filter passed somatic variants and detected about 150 to 600 filtered mutations in the first deep sequencing but about 50 to 120 filtered mutations in second (Figure 3A). In Mutect2, the detected variant number in the second deep sequencing was more than in the first, but in LoFreq, the number in the first deep sequencing was more than in the second. Using Strelka2 the number was similar between the first and second deep sequencing. These features were independent of the three different polymerases and were constant for each variant caller. These results indicate that each of these three somatic variant callers have different characteristics for variant detection.

We next compared somatic variants detected using the three different DNA polymerases for each deep sequencing experiment (Figure 3B). Most somatic variants were DNA polymerase-specific variants (Group 1, 2, and 3). Although the number of filtered variants were reduced in Group 1, 2, and 3, the ratio of Group 1, 2, and 3 to all variants was increased (Figure 3C). We also compared somatic variants detected using three different somatic variant callers for each deep sequencing experiment (Figure 3D). Almost all variants detected using LoFreq were also included in the variants detected using Strelka2 (Group 6 and 7). More than 75% of the variants detected using Mutect2 were included in the variants detected using Strelka2 (Group 4 and 7). Among the filtered variants, the Strelka2-specific variants (Group 1) were reduced. However, the ratio of DNA polymerase-specific variants (Group 1, 2, and 3) did not change, since Mutect2-specific variants (Group 2) were increased after the filtering process (Figure 3E). These results suggest that most somatic variant calls are due to polymerase errors during PCR, i.e., false positives due to inaccurate base incorporation in the initial amplification cycle and/or false negatives generated by preferential amplification of one allele, those based on the nature of the polymerase and the DNA sequence. The ability of variant callers to distinguish between real somatic variants and artifacts is not sufficient; the detected variants from each experiment were relatively similar even if different variant callers were used.

Next, we compared the number of somatic variants detected in duplicate deep sequencing and categorized the somatic variants into four groups. The groups are as follows: category 1, variants detected using all three polymerases in both of the sequencing experiments; category 2, variants detected in two of three polymerases in both of the sequencing experiments; category 3, variants detected in two of three polymerases in a single sequencing experiment only; category 4, variants detected using only one polymerase. Surprisingly, there were only five category 1 and category 2 variants when using both Strelka2 and Mutect2, and only two using LoFreq (Figure 3F). The percentage of category 1 and category 2 variants was less than 1% of all detected variants, regardless of the somatic variant caller (Figure 3G). However, these category 1 and category 2 variants were detected with high overlap in the three variant callers (Figure 3H). The number of filtered variants in category 1 and 2 are fewer than the non-filtered variants when using Strelka2 and Mutect2, although the ratio of filtered variants in category 1 and 2 is higher than non-filtered when using Strelka2 (Figure 3G). These results indicate that most of the experiment-specific variants were generated at the PCR step due to random polymerase errors. However, focusing on common variants using dual deep sequencing effectively eliminates these errors. Using three somatic variant callers, we found that these programs could detect similar variants from same sequencing data. Filtering processing increases the ratio of positive variants, but decreases the number of positive variants.

### 2.3. Validation of Detected Variants by CHIPS Technology and Sanger Sequencing

To examine the accuracy of six common category 1 or 2 variants, including variants that passed and did not pass the filter, we validated these variants using CHIPS technology and Sanger sequencing (Figure 4A). Five of the six variants contained cleaved bands in CEL nuclease-treated tumor samples (Figure 4B). Sanger sequencing showed somatic variant peaks in the tumor samples, as well as in the Integrative Genomic Viewer (IGV) (Figure 4C–E). On the other hand, the 9 variants in category 3 that passed filtering were not confirmed by CHIPS and Sanger sequencing (Figure 4F). The ratio of category 1 or 2 variants among validated variants is highest using Strelka2, followed by Mutect2, with the lowest in LoFreq (Figure 4A). These results indicate that it is possible to identify real somatic mutations with a high probability using dual deep sequencing; different somatic variant tools change the number and accuracy of detected mutations.

### 2.4. Comparison of Variant Allele Frequency between Dual Deep Sequencing

We compared the allele frequency of mutations detected using dual deep sequencing with three different DNA polymerases (Figure 5A). The average variant allele frequencies were from 0.15 to 0.3 with a standard deviation of 0.05 to 0.11. We compared the variant allele frequencies using three different variant callers (Figure 5B) and between dual deep sequencing with three different DNA polymerases (Figure 5C). Although the differences in allele frequency between the three different polymerases and duplicate experiments for each variant were large, with a wide range of distribution, the fluctuation widths were similar between three variant callers (Figure 5B). On the other hand, the variant allele frequencies determined by the three variant callers were almost same and were distributed in a narrow range in each deep sequencing for any polymerase (Figure 5C). These results indicate that the variance of variant allele frequency is generated at the PCR step of each experiment and not at the somatic variant calling. All DNA polymerases had poor reproducibility in variant allele frequency. The differences in variant allele frequency between DNA polymerases are large, but the differences between experiments are larger than between DNA polymerases and somatic variant callers. Therefore, we hypothesize that the main cause of the fluctuations in variant allele frequency is due to the variation in uniformity of target amplification between alleles by PCR. There was no difference among the three polymerases in this regard.

### 2.5. Annotation of Validated Somatic Variants

To assess which is the pathogenic variant, we annotated the validated somatic variants using clinical databases (dbSNP, ClinVar and COSMIC) (Table 1). *FBXW7* (chr4:152326214; C > T, NM_033632.3:c.1436G > A p.(Arg479Gln)) and *CDKN2A* (chr9:21974793; G > A, NM_058197.4:c.35C > T p.(Ser12Leu)) are registered in dbSNP, ClinVar, and COSMIC. *MAP3K1* (chr5: 56882328; C > T, NM_005921.1: c.3128C > T p.(Ser1043Phe)) is registered only in COSMIC. We removed other variants that were intron variants without pathogenic information. In silico analysis showed that *FBXW7* and *MAP3K1* variants had a high score (damaging) in most databases, but *CDKN2A* was considered benign (Table 2). In the COSMIC database, the same *FBXW7* mutation was found in cervical squamous cell carcinoma and endometrioid carcinoma, and the same *MAP3K1* mutation was found in cutaneous squamous cell carcinoma. The variant allele frequencies of *FBXW7* (chr4: 152326214) and *MAP3K1* (chr5: 56882328) are 31.6% ± 11.0% and 16.4% ± 4.6% respectively (Figure 5A). It seems that both alleles in the tumor have the *FBXW7* mutation because the tumor cell content was estimated to be 30–40%. In contrast, the *MAP3K1* mutation, which is a loss-of-function mutation, seems to exist in one allele, and the mutation status of the other allele is unknown. It is reported that FBXW7 targets mTOR for degradation and cooperates with PTEN in tumor suppression [23]. Therefore, it is conceivable that a loss-of-function mutation with LOH of *FBXW7* may cause overactivation of the mTOR pathway and promote tumor growth. These results suggest that the *FBXW7* mutation may play an important role in this cervical cancer.

### 2.6. Evaluation of Read Number and Different DNA Polymerases

We extracted any number of reads of samples using the Seqtk software to evaluate the status between different read numbers of the different DNA polymerases. The coverage rate increased from 50,000 reads to 300,000 reads but did not change from 300,000 reads to 600,000 reads (Figure 6A). The coverage rate of tumor samples was lower than in blood samples. However, there were no differences in coverage rates between the different DNA polymerases. The on target rate of tumor samples was lower than blood samples with no differences between the polymerases. The on target rate decreased gradually from 50,000 reads to 600,000 reads (Figure 6B). The percentage of 100% coverage target regions increased, but was moderate from 300,000 reads (Figure 6C). Similarly, the percentage of 0% coverage target regions decreased, but was moderate from 300,000 reads (Figure 6D). On the other hand, the average depth on target regions increased without reaching a plateau as the number of reads was increased (Figure 6E). We next examined the accuracy and the detection ability of three somatic variant callers (Strelka2, Mutect2, and LoFreq) in each read number (Figure 6F,G). The variant number detected by Strelka2 and Mutect2 increased gradually, but the number detected by LoFreq did not change when increasing read numbers from 200,000 to 600,000 (Figure 6F). In Strelka2, the detected number of category 1 and 2 variants reached the maximum detection number at 400,000 reads (Figure 6G). On the other hand, the detected variants using Mutect2 and LoFreq did not yet reach a maximum at 600,000 reads. These results indicate that Strelka2 need fewer reads for the detection of accurate somatic mutations as compared with Mutect2 and LoFreq.

## 3. Discussion

Many high-fidelity DNA polymerases are used for library amplification, and various somatic variant callers and pipelines have been developed to detect somatic mutations in clinical genetic testing. Several studies have compared the fidelity of DNA polymerases and the performance of somatic variant callers and pipelines [24,25,26,27,28,29,30]. However, to date, the reproducibility of somatic variant calling has not been assessed. Therefore, we performed duplicate deep sequencing of the same samples to compare the accuracy of DNA polymerases and somatic variant callers.

Of the high-fidelity DNA polymerases used in this study, the KAPA and NEB polymerases showed higher coverage rates and average depths than the Agilent polymerase. However, the differences between duplicate sequencing were greater than differences in amplification performance between DNA polymerases. Furthermore, the differences in allele frequency of somatic mutations were more dependent on PCR reactions of each sequencing experiment than on the difference between the DNA polymerases. Moreover, almost all detected variants are polymerase-specific in both duplicate sequencing experiments, even if the same template was used for library pre-amplification. High sequencing depth merely generated false positives using the Illumina sequencing technology. These false positives included several ClinVar-reported pathogenic variants as well as many nonsense and frame shift variants. Therefore, these false positives can affect the wrong direction in the clinical interpretation of the results. These results suggest that deep sequencing must be performed in at least duplicate for detection of accurate somatic mutations.

Using the Strelka2 somatic variant caller detected more somatic variants than using Mutect2 and LoFreq, although most variants detected in Strelka2 were false-positive. The filtering process that is provided using Strelka2, Mutect2, and LoFreq reduced not only the number of false-positive variants but also true-positive variants. Using duplicate experiments and taking the common variants between the experiments was more effective at eliminating false-positive variants than using the filtering programs of the variant callers. Strelka2 needed fewer reads for the detection of somatic variants than Mutect2 and LoFreq. The numbers of variants detected using Mutect2 and LoFreq, but not Strelka2, were strongly affected by differences in the performances between duplicate deep sequencing. These results suggest that Strelka2 is better than Mutect2 and LoFreq in somatic variant calling with dual deep sequencing. Most detected variants in the first sequencing experiment were not detected in the second sequencing experiment. Even if Strelka2 was used, a few validated real mutations were not detected using some of the DNA polymerases in a single sequencing experiment. These results suggest that single sequencing is not sufficient for accurate detection.

Several studies suggest that PCR errors in library preparation cause inaccurate variant allele frequencies [10,11,12,13]. In this study, variant allele frequency was not a coincidence due to random fluctuations of amplification efficiency between alleles during the PCR step of each library preparation. The differences of variant allele frequency were found in any DNA polymerases and the differences of variant allele frequency between each sequencing with the same DNA polymerase was greater than the difference between somatic variant callers and/or DNA polymerases. These results suggest that multiple experiments are required to determine more accurate variant allele frequency.

This study demonstrates that deep sequencing should be performed at least in duplicate to accurately detect low-frequency somatic mutations and correctly determine allele frequencies. We also demonstrate that Strelka2 is better than Mutect2 and LoFreq in somatic variant calling. Pooling pre-capture libraries amplified independently by several DNA polymerases with different indexes is a cost-effective and useful method to accurately detect low-frequency somatic mutations.

## 4. Materials and Methods

### 4.1. Patient and Sample

The patient participating in this study was a 54-year-old female with cervical cancer. This patient underwent chemoradiotherapy, extended hysterectomy, and extirpation of bilateral appendages. Pathological examination revealed squamous cell carcinoma with relatively high levels of stromal cells. The tumor cell content of the specimen was estimated to be 30–40%. Written informed consent was obtained from the patient and the study design was approved by the ethics review board of Kanazawa Medical University (reference number G138).

### 4.2. Genomic DNA Extraction

A total of 20 serial sections of 1 square centimeter with 5 μm thickness were cut from formalin-fixed paraffin-embedded (FFPE) cervical cancer samples. Genomic DNA was extracted using the GeneRead DNA FFPE Kit (Qiagen, Hilden, Germany) according to the manufacturer’s standard protocol. Blood DNA was also extracted by QIAamp DNA Mini Kit (Qiagen) for germline controls.

### 4.3. Library Preparation

The genomic DNA was fragmented using Covaris M220 to achieve a target peak of 300 bp (duty factor, 20.0%; peak incident power, 50.0; cycles per burst, 200; duration, 80 s; Covaris, Woburn, MA). Agilent SureSelect XT HS Target Enrichment protocol (Agilent Technologies, Santa Clara, CA) was followed for library preparation with the modification at the pre-amplification step. After adapter ligation, the adapter-ligated DNA fragments are divided and amplified (14 cycles) by three different high-fidelity DNA polymerases (Herculase II fusion DNA polymerase (Agilent), KAPA HiFi HotStart DNA polymerase (Kapa Biosystems, Wilmington, MA), and Q5 High-Fidelity DNA polymerase (New England Biolabs, Ipswich, MA)). Pooled pre-amp libraries were hybridized to the cancer gene-targeted capture probes (SureSelect NCC oncopanel, Agilent) and the hybridized pre-libraries were captured using streptavidin-coated beads. The captured libraries were then amplified (12 cycles). The quality of the libraries was assessed using the TapeStation 4200 with the High Sensitivity D1000 ScreenTape (Agilent).

### 4.4. Sequencing and Generation of FASTQ Files

Libraries were quantified using HS Qubit dsDNA assay (Thermo Fisher Scientific, Waltham, MA) and were sequenced on the Illumina MiSeq (2 × 150 bp) following standard Illumina protocol (Illumina, San Diego, CA). The FASTQ files containing the molecular barcodes were generated using the bcl2fastq software (Illumina).

### 4.5. Data Analysis

The FASTQ files were trimmed for quality using the SureCallTrimmer software (Agilent) and the trimmed FASTQ files were aligned to the reference human genome (hg38) using the Burrows-Wheeler Aligner MEM algorithm (BWA-MEM version 0.7.17-r1188) [31]. Duplicate reads were removed using the Agilent Genomics NextGen Toolkit (AGeNT) software (Agilent) according to the molecular barcode information. Somatic mutations were identified using Strelka2 (Version 2.9.10) [16], GATK’s Mutect2 (Version 4.0.6.0), and LoFreq (Version 2.1.3.1), separately [17,18]. For analysis and interpretation, we used the following software packages: SAMtools (Version 1.9), BEDTools (Version v2.27.1), BCFtools (Version 1.9), vcftools (Version 0.1.16), Seqtk (Version 1.3-r106), and Integrative Genomic Viewer (IGV Version 2.4.13) [32,33,34,35,36]. For variant annotation, we used the Database of Short Genetic Variations dbSNP (Version 151), ClinVar, and COSMIC (Version 87) [37,38,39]. For in silico analysis, we used the database dbNSFP (Version 3.2) that compiles prediction scores from 29 prediction algorithms [40]. The raw data in this study have been deposited in the Sequence Read Archive database of NCBI under the BioProject accession number PRJNA629636.

### 4.6. CHIPS and Sanger Sequencing

To verify the detected somatic variants, site-specific PCR primers were designed using Primer3 and CEL nuclease-mediated heteroduplex incision with polyacrylamide gel electrophoresis and silver staining (CHIPS) analysis and Sanger sequencing were performed [41]. CHIPS is a highly sensitive mutation screening method based on enzyme mismatch cleavage and was performed as described previously [42,43,44]. Direct DNA sequencing was performed using the BigDye Terminator v3.1 cycle sequencing kit and ABI PRISM 3100xl Genetic analyzer (Thermo Fisher Scientific).

## 5. Conclusions

In summary, the differences between detected variants in duplicate sequencing were greater than the differences between DNA polymerases and somatic variant callers in single sequencing. Furthermore, the differences in allele frequency of real somatic mutations were more variable between duplicate sequencing than between DNA polymerases and somatic variant callers. The detected variants in Strelka2 are more than in Mutect2 and LoFreq. Although the detected variants are almost false-positive, the common variants in duplicate sequencing are positive mutations. We conclude that deep sequencing should be performed at least in duplicate to detect accurate somatic mutations and determine accurate allele frequencies.

## Figures and Tables

**Figure 1 ijms-21-03530-f001:**
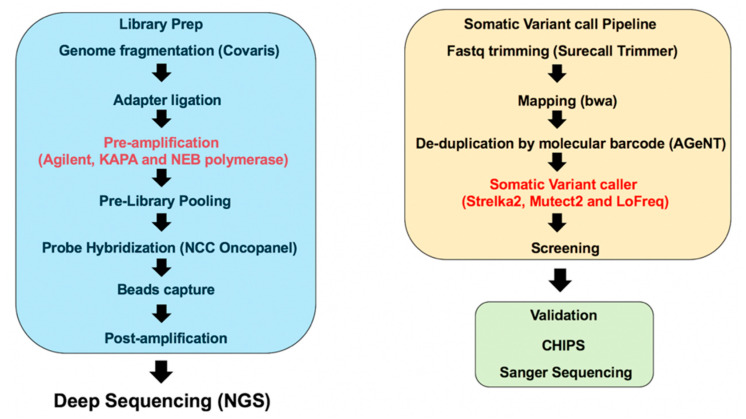
Workflow for somatic variant calling with next generation sequencing (NGS). During the library prep, we used three different DNA polymerases for pre-amplification to compare the fidelity of these polymerases. The sequencing data are the result of the somatic variant calling pipeline. We used three different somatic variant callers to compare the detected number of variants and accuracy of variant detection. Detected variants were validated using CHIPS and Sanger sequencing.

**Figure 2 ijms-21-03530-f002:**
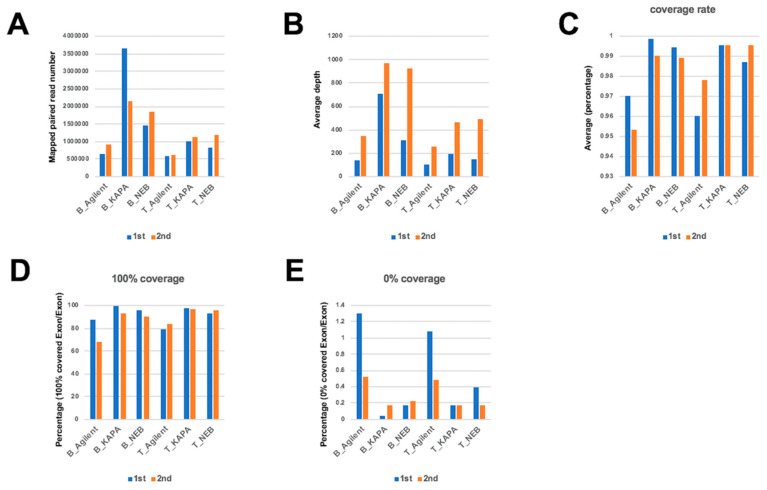
Comparison for dual deep sequencing. (**A**) The mapped read number for each sample. (**B**) The average depth was plotted for each polymerase in dual sequencing. (**C**) The average coverage rate percentages. (**D**) The percent of 100% coverage target regions. (**E**) The percentage of 0% coverage target regions. B: Blood sample T: Tumor sample

**Figure 3 ijms-21-03530-f003:**
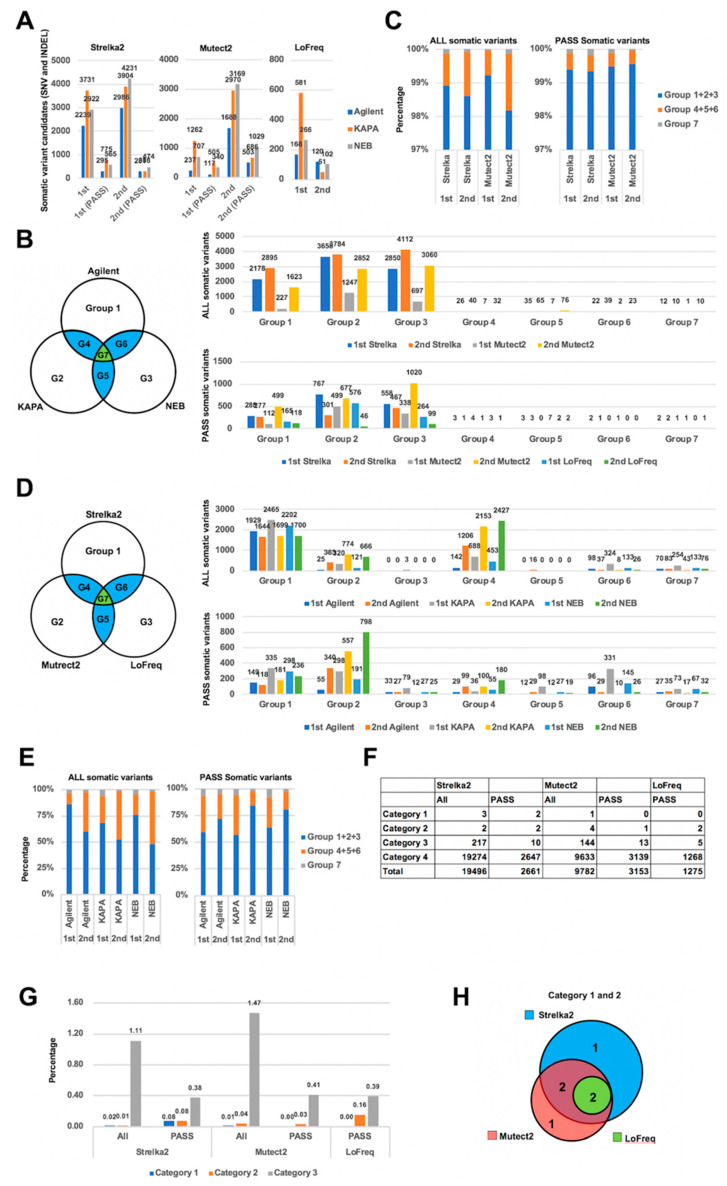
Comparison of the somatic variant callers in dual deep sequencing. (**A**) The number of somatic variants called by each somatic variant caller. (**B**,**D**) Venn diagrams showing the groups of the detected somatic variants using three different DNA polymerases with each of the somatic variant callers (**B**) and the three somatic variant callers with each of the DNA polymerases (**D**). The number of all somatic variants (top) and passed somatic variants (bottom) in each group. (**C**) The ratio of variant groups in Figure 3B, variants detected in only one polymerase (Group 1, 2, and 3), variants common to two polymerases (Group 4, 5, and 6) and variants common to all polymerases (Group 7), were plotted for each variant caller and each experiment. (**E**) Similarly, the ratio of variant groups by the number of variant callers detecting the same variant in Figure 3D were plotted for each polymerase and each experiment. (**F**) Classification of the variants into 4 categories. (**G**) The percentage of variants in category 1, 2, and 3. (**H**) Venn diagram showing the number of somatic variants that are detected in category 1 and 2.

**Figure 4 ijms-21-03530-f004:**
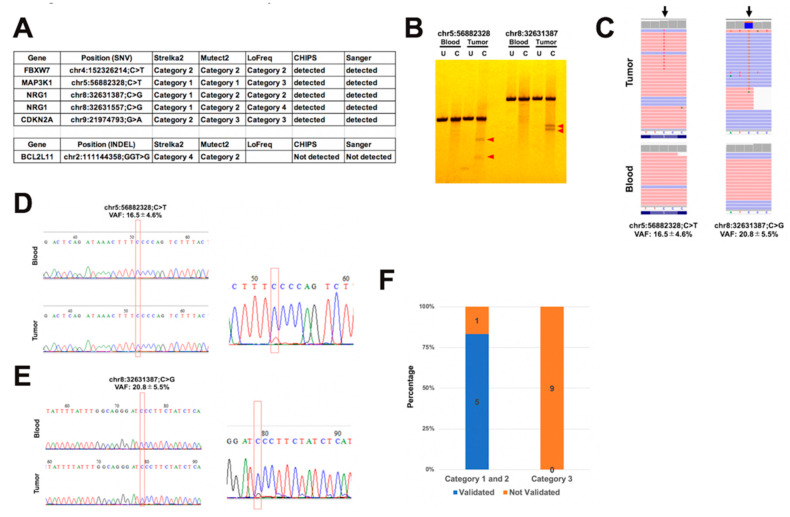
Validation for the detected somatic variants by CHIPS and Sanger sequencing. (**A**) Summary of validation results. (**B**) CHIPS technology assay. Red arrow head shows cleaved heteroduplexes. U: uncut C: cut by CEL nuclease. (**C**) Aligned reads visualized by Integrative Genomic Viewer (IGV) at the detected somatic variant positions. (**D**) and (**E**) Electropherograms of Sanger sequencing of the detected somatic variants in blood and tumor samples. (**F**) Ratio of validated and not validated variants in category 1, 2, and 3.

**Figure 5 ijms-21-03530-f005:**
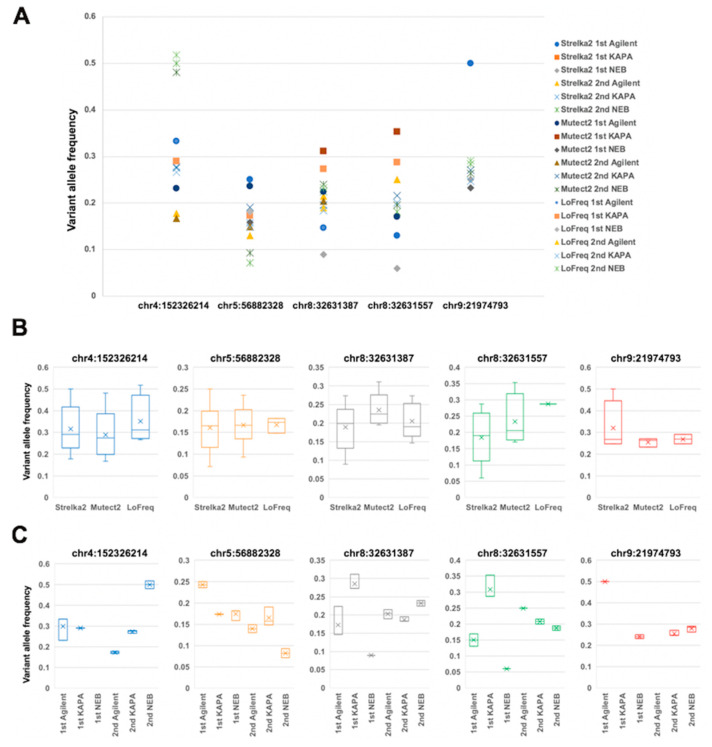
Comparison of variant allele frequency between dual deep sequencing. (**A**) Variant allele frequency of detected variants for each somatic variant caller in each sequencing experiment with the three DNA polymerases. (**B**) Variant allele frequency for each somatic variant caller. (**C**) Variant allele frequency for each sequencing experiment with the three DNA polymerases.

**Figure 6 ijms-21-03530-f006:**
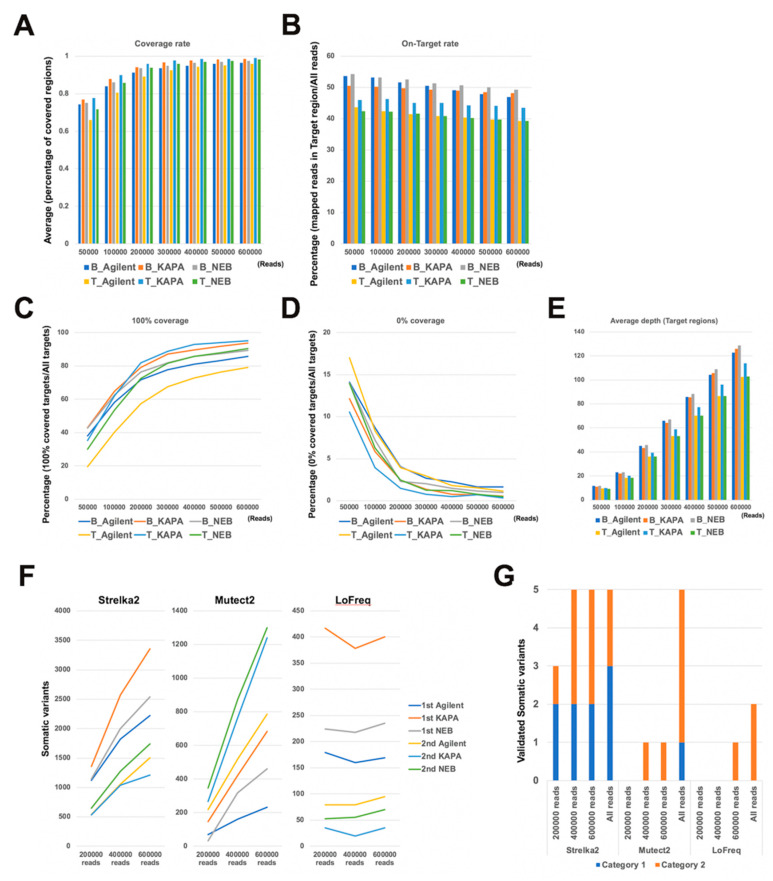
Effect of read number and different DNA polymerases. (**A**) The average coverage rate percentage plotted for each read number. (**B**) The percent of mapped reads on the target region plotted for each read number. (**C**) The percentage of 100% coverage target regions plotted for each read number. (**D**) The percentage of 0% coverage target regions plotted by each read number. (**E**) The average depth plotted for each read number. (**F**) The number of variants detected by somatic variant callers plotted for each read number. (**G**) The number of accurate mutations detected by each somatic variant caller plotted for each read number. B: Blood sample T: Tumor sample.

**Table 1 ijms-21-03530-t001:** Summary of the validated mutations.

Gene	Position (hg38)	DNA	Mutation Type	Protein	dbSNP	ClinVar	COSMIC
FBXW7	chr4:152326214	NM_033632.3:c.1436G > A	missense	NP_361014.1:p.Arg479Gln	rs866987936	376419	COSM1154291; COSM22974; COSM447498; COSM447499; COSM6847976: COSM94297
MAP3K1	chr5:56882328	NM005921.1:c.3138C > T	missense	NP_005912.1:p.Ser1043Phe	-	-	COSM6889390
NRG1	chr8:32631387	NM_13960.4:c.502+14502C > G	intron variant	-	-	-	-
NRG1	chr8:32631557	NM_13960.4:c.502+14672C > G	intron variant	-	-	-	-
CDKN2A	chr9:21974793	NM_058197.4:c.35C > T	missense	NP_478104.2:p.Ser12Leu	rs141798398	236988	COSM6985693; COSM6985694; COSM6985695

**Table 2 ijms-21-03530-t002:** Summary of the validated mutations in silico analysis.

Gene	Position	Mutation type	Protein	SIFT	Polyphen2_HDIV	Polyphen2_HVAR	LRT	Mutation Taster	MutationAssessor	FATHMM	PROVEAN	MetaSVM	MetaLR	fathmm-MKL_coding
FBXW7	chr4:152326214;C > T	missense	p.Arg479Gln	Damaging	Probably damaging	Probably damaging	Deleterious	Disease causing	Low	Tolerated	Deleterious	Tolerated	Tolerated	Deleterious
MAP3K1	chr5:56882328;C > T	missense	p.Ser1043Phe	Damaging	Probably damaging	Probably damaging	Deleterious	Disease causing	Medium	Tolerated	Neutral	Deleterious	Deleterious	Deleterious
CDKN2A	chr9:21974793;G > A	missense	p.Ser12Leu	Tolerated	Benign	Benign	.	Polymorphism	Neutral	Tolerated	Neutral	Tolerated	Tolerated	Neutral

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
