# Peer review of "Dual Deep Sequencing Improves the Accuracy of Low-Frequency Somatic Mutation Detection in Cancer Gene Panel Testing"

_ijms, 2020, doi:10.3390/ijms21103530_

Round 1

Reviewer 1 Report

I found this article to be well-written and informative, with good data presentation and discussion.  The abstract provided a thorough and accurate summary about the experiments performed.  My only issue was the statement in the results section, with respect to the variant allele frequency, that the variant allele frequency "is not correct due to random fluctuations of amplification efficiency for specific targets during the PCR step of each library preparation, and differences of variant allele frequency between each sequencing with same DNA polymerase was greater than difference between somatic variant caller of DNA polymerases."  and "Most detected variants in the first sequencing experiment were not detected in the second sequencing experiment."  This was confusing, as how was the VAF not correct? This is not clarified. It is also a bit confusing when it is concluded that Strelka2 is the best variant caller, but "some validated real mutations were not detected using some of the DNA polymerases in single sequencing experiment."  How are you able to determine Strelka2 is the best caller for this variable if the variant allele frequency was not correct? I believe some additional clarification or explanation might be warranted.  

Author Response

Dear Reviewer,

Thank you for giving us the opportunity to submit a revised draft of our manuscript titled “Dual deep sequencing improves the accuracy of low-frequency somatic mutation detection in cancer gene panel testing” to the International Journal of Molecular Sciences. We appreciate the time and effort that you have dedicated to providing your valuable feedback on our manuscript. We are grateful to you for your insightful comments on our paper. We have been able to incorporate changes to reflect all of your suggestions. We highlighted the changes within the manuscript.

Here is a point-by point response to your comments and concerns.

Point 1: the variant allele frequency "is not correct due to random fluctuations of amplification efficiency for specific targets during the PCR step of each library preparation, and differences of variant allele frequency between each sequencing with same DNA polymerase was greater than difference between somatic variant caller of DNA polymerases."  and "Most detected variants in the first sequencing experiment were not detected in the second sequencing experiment."  This was confusing, as how was the VAF not correct?

Response 1: We have added several references and revised to emphasize this point. Several studies (reference 11-13) showed that the variant allele frequency is variable between experiments and suggested the variance was caused by PCR error. In this study, the variance of variant allele frequency was less between DNA polymerases and/or somatic variant callers than between experiments. These results suggest that the differences of variant allele frequency were mainly caused by random PCR error that cause un-uniformity of target amplification between alleles, in each experiment (line 51-52 and 54-55 in page 2, line 213-217 in page 8, line 312-318 in page 12).  

Point 2: Strelka2 is the best variant caller, but "some validated real mutations were not detected using some of the DNA polymerases in single sequencing experiment."  How are you able to determine Strelka2 is the best caller for this variable if the variant allele frequency was not correct?

Response 2: We have revised to emphasize this point. Strelka2 were better than other two somatic variant callers. Even if Strelka2 was used, a few validated real mutations were not detected using some of the DNA polymerases in single sequencing experiment (line 307-311 in page 12).

Reviewer 2 Report

This manuscript studies various variant callers as it relates to cancer variant detection and conclude that the Strelka2 analysis was superior.

The manuscript is well written, and findings are notable.  Some additional areas to highlight.

A more comprehensive review of the available somatic variant callers (ref 15-18) and why Strelka, Mutect2, and LoFreq were chosen to be compared.

Also, were the variants that were discrepant of any clinical significance?

Author Response

Dear Reviewer,

Thank you for giving us the opportunity to submit a revised draft of our manuscript titled “Dual deep sequencing improves the accuracy of low-frequency somatic mutation detection in cancer gene panel testing” to the International Journal of Molecular Sciences. We appreciate the time and effort that you have dedicated to providing your valuable feedback on our manuscript. We are grateful to you for your insightful comments on our paper. We have been able to incorporate changes to reflect all of your suggestions. We highlighted the changes within the manuscript.

Here is a point-by point response to your comments and concerns.

Point 1: A more comprehensive review of the available somatic variant callers (ref 15-18) and why Strelka, Mutect2, and LoFreq were chosen to be compared.

Response 1: We have revised to emphasize this point. We chose Strelka2, Mutect2 and LoFreq because several studies already reported that these variant callers are relatively accurate and reliable than other somatic variant callers. We focused on comparison of utility and stability of these somatic variant callers in dual deep sequencing (line 62, 63, 65 and 66 in page 2).

Point 2: Also, were the variants that were discrepant of any clinical significance?

Response 2: We have revised to emphasize this point. The false-positive variants included several ClinVar-reported pathogenic variants as well as many nonsense and frame shift variants. These variants may affect discrepant of clinical significance (line 294-296 in page 12).
